# Triple Priority: TB/HIV Co-Infection and Treatment Outcomes among Key Populations in The Kyrgyz Republic: A National Cohort Study (2018–2022)

**DOI:** 10.3390/tropicalmed8070342

**Published:** 2023-06-27

**Authors:** Olga Goncharova, Arpine Abrahamyan, Divya Nair, Mher Beglaryan, Aibek Bekbolotov, Elena Zhdanova, Abdullaat Kadyrov, Rony Zachariah

**Affiliations:** 1National Center of Phthisiology, Bishkek 720000, Kyrgyzstan; zhdanova-ev@mail.ru (E.Z.); abdylat.kadyrov@gmail.com (A.K.); 2Tuberculosis Research and Prevention Centre, Yerevan 0014, Armenia; arpine_abrahamyan@yahoo.com (A.A.); mherbeglaryan@gmail.com (M.B.); 3International Union Against Tuberculosis and Lung Disease, 75001 Paris, France; divya.nair@theunion.org; 4The Republic Center of AIDS, Bishkek 72000, Kyrgyzstan; aibek_0001@mail.ru; 5United Nations Children Fund, United Nations Development Programme, World Bank, World Health Organization, Special Programme for Research and Training in Tropical Diseases (TDR), 20, Avenue Appia, CH-1211 Geneva, Switzerland; zachariahr@who.int

**Keywords:** health systems strengthening, universal health coverage, SDGs, SORT IT, ERI-TB, end-TB, vulnerable populations, key populations

## Abstract

**Background**: Improving tuberculosis (TB) care in key populations is an operational research priority in the Kyrgyz Republic. Here, we describe the characteristics of TB/HIV co-infected individuals, their affiliations with key country-wide population groups, and their TB treatment outcomes. **Methods:** This was a cohort study using national programmatic data (2018–2022). The key population groups included people with increased exposure to TB, limited access to TB services, and increased risk of acquiring TB. **Results:** Among 693 individuals with TB/HIV co-infection, the majority (58%) of individuals were from two regions of the Kyrgyz Republic (Chui and Bishkek). Eighty-four percent (84%) individuals had one or more affiliations to eight key population groups, with 49% of the individuals affiliated to ≥2 groups and 92% of the individuals were on both antiretroviral treatment and cotrimoxazole preventive therapy. Overall, 406 (59%) of the individuals had successful outcomes and 287 (41%) of the individuals had unsuccessful outcomes. Unsuccessful outcomes increased from 36% (n-39) with TB/HIV alone to 47% (n-86) with affiliations to ≥3 key population groups (*P*-0.03). Unsuccessful outcomes were associated with co-morbidities (diabetes mellitus and hepatitis B/C), migration, alcohol use, and extrapulmonary TB. **Conclusions:** For a long time, people with TB/HIV co-infection have been recognized as a “double priority”. Affiliation to key populations accentuates their status to “triple priority”. We advocate for increased attention and equity towards these populations.

## 1. Introduction

Globally, in 2021, 40% of the estimated 10.6 million people with tuberculosis (TB) were missed by health systems (known as “missed TB cases”). In the Kyrgyz Republic, this figure was estimated to be 46% [1]. Missed TB cases have been shown to be disproportionally concentrated among key populations [2,3,4].

Key populations in the context of TB include those who have increased exposure to TB due to where they live or work (e.g., prisoners, health workers, and homeless); people who have limited access to quality TB services (e.g., undocumented migrants, refugees, or those internally displaced); people at increased risk to acquire TB because of clinical or behavioral factors that compromise immune function (e.g., people living with human immunodeficiency virus (HIV), intravenous drug users, alcohol overuse, and diabetes mellitus) [4].

From a health system perspective, key populations are hard to reach and underserved since they are marginalized and/or criminalized and, often, the last to be reached by passive case-finding approaches that rely on individuals presenting by themselves to health facilities. Since key populations constitute an underserved group, ensuring effective case finding and treatment is of utmost importance to public health.

To attain the World Health Organization (WHO) end TB targets of 90% reduction in TB deaths and 80% reduction in TB incidence by 2030, a focus on key population groups is urgently needed [5]. Two of the three targets in the end TB strategy involve key populations. One of these targets is to ensure that TB diagnostic and treatment services reach at least 90% of key populations, while the other is to achieve a minimum of 90% treatment success [5].

The Kyrgyz Republic is one of the high priority countries for multidrug-resistant TB (MDR-TB) and the National Tuberculosis Programme (NTP) has a specific focus on key populations [1]. The TB National Data Registry compiles country-wide data on their clinical characteristics and treatment outcomes.

A PubMed search revealed two previous studies in key populations that used data from the National TB Registry of the Kyrgyz Republic. One study was restricted to migrants [6] and the other study focused on one region of the country [7]. To date, there has been no country-wide assessment on who these population groups are, where they come from, and how they fare on TB treatment. The latter is particularly important since the management of TB in those who are also co-infected with HIV is already challenging, but when key populations are added to this picture the management is further complicated by social and behavioral factors and the potential need for multiple treatment regimens (triple priority).

Performing such an analysis with an operational research lens could provide useful intelligence for informed decision making. Thus, in this study, we aimed to describe the characteristics of TB/HIV co-infected individuals from key population groups and their TB treatment outcomes in the Kyrgyz Republic. Among TB/HIV co-infected individuals, the specific objectives were, for the period from January 2018 to June 2022, to determine the following:Sociodemographic and clinical characteristics and the composition of the key population groups;TB treatment outcomes stratified by key population groups and types of drug resistance;Factors associated with unsuccessful treatment outcomes.

## 2. Materials and Methods

### 2.1. Study Design

This cohort study used routinely collected secondary programmatic data from the Kyrgyz Republic’s National TB Registry.

### 2.2. Study Setting

The Kyrgyz Republic is located in Central Asia, bordered by China, Kazakhstan, Tajikistan, and Uzbekistan. The population is around 7 million, and the country has seven regions, with Bishkek as the capital [8]. The proportion of unemployment is around 9%, and 22% of the population live under the poverty line [9].

### 2.3. Key Population Groups

Key populations in the context of the Kyrgyz Republic’s National TB Registry include all categories included under the variable “TB risk group”. These include migrants, internally displaced, homeless, drug users, alcohol users, unemployed, prisoners/ex-prisoners, those with comorbidities (HIV, diabetes mellitus and hepatitis B/C), health workers, and TB contacts. For the purposes of this study, we referred to them as key population groups. TB “migrants” are defined as those diagnosed with TB within 6 months of arrival in a new region (internal migration) or within 3 months of arrival in the Kyrgyz Republic after 3 months stay abroad (external migration). “Homeless” are defined as people with no identity cards and deemed to have no permanent residence. The case-finding strategy was passive but key population groups were informed about TB symptoms and that they should present themselves to health services.

### 2.4. Management of TB and HIV

TB management follows the WHO guidelines [10]. TB care is provided through a network of TB diagnostic and treatment facilities, which include TB hospitals and family medicine centers (primary polyclinic services) [11,12].

Once a patient is confirmed with TB, he/she is referred to a TB physician for registration and for the start of treatment. Drug-sensitive and drug-resistant TB cases receive standardized TB treatment regimens after sputum specimens are taken for mycobacterial culture and drug sensitivity testing (DST). The Xpert^®^ MTB/RIF assay (Cepheid Inc, Sunnyvale, CA, USA) is used in this country. The results of an Xpert^®^ MTB/RIF assay is available in two hours, and if rifampicin resistance is diagnosed, the patient is started on standard MDR-TB treatment (DOTS-plus) while awaiting the DST results. The TB regimens include individualized or short regimens in line with the WHO guidelines [10]. When the results arrive in 6 weeks, the treatment regimen is tailored (individualized). Anti-tuberculosis treatment is offered free of charge for the residents, while non-residents are required to be covered by insurance schemes.

HIV testing is offered to all TB patients, and patients who are found to be HIV positive are offered cotrimoxazole preventive therapy and are eligible for antiretroviral treatment (ART) which is started as soon as possible [13]. HIV management is offered in separate HIV clinics and under the aegis of the National HIV Programme. ART regimens are standardized according to national and WHO guidelines [13,14]. The most common ART regimen is a combination of tenofovir with lopinavir and dolutegravir.

### 2.5. TB Treatment Outcomes

TB treatment outcomes are standardized according to the WHO guidelines (Table 1). For the purposes of this study, we categorized successful treatment outcomes as the combination of cured and treatment completed, while unsuccessful outcomes included death, treatment failures, lost to follow-up, and not evaluated. Since some individuals experienced multiple episodes of TB treatment during the study period, only treatment outcomes of the first episode of TB were reported in this study.

### 2.6. Study Population and Period

The study population consisted of all TB/HIV co-infected individuals from any key population group(s) and registered by the NTP between January 2018 and June 2022. If an individual experienced multiple episodes of TB during this period, then, only the first episode was considered for analysis.

### 2.7. Data Sources and Variables

Data on sociodemographic and clinical characteristics of key populations were obtained from the TB master cards and the TB treatment registers. Data in the master cards were cross verified with the hospital TB treatment registers, and then entered into Microsoft Excel (v. 2010, Microsoft Corp, Redmond, WA, USA).

Data variables included a unique patient identifier, date of TB registration, age, sex, region, type and category of TB, type of drug resistance, ART, date of starting ART, cotrimoxazole preventive therapy (CPT), belonging to key population group, and TB treatment outcome.

### 2.8. Statistical Analysis

The results were presented using descriptive statistics (numbers and percentages), tables, and figures. The country-wide geographic distribution of key populations is presented graphically by regions. For age categories, we opted for a context-specific age grouping. In the Kyrgyz Republic, 18 years is the legal age for seeking employment and 55 years is the usual age of retirement. Between 18 and 55 years, individuals are engaged in economically productive activities. Measures of risk were estimated using relative risks (RR) and adjusted using log binomial regression. All available variables were included in the initial regression model. Since individuals may belong to multiple key population groups (e.g., homeless and intravenous drug user), we also categorized individuals into those belonging to 1, 2, or ≥3 key population groups. The dependent variable for risk factor analysis was “unsuccessful TB treatment outcome”. Variables were adjusted using log binomial regression as this was an explanatory model; we included all the variables and did not conduct any diagnostic tests which could be appropriate for predictive models.

A *p*-value of <0.05 was considered to be statistically significant. The descriptive analysis was performed using the EpiData Analysis software (version 2.2.2.182, EpiData Association, Odense, Denmark) and a log binomial regression analysis using Stata v. 14 (StataCorp, College Station, TX, USA).

## 3. Results

### 3.1. Sociodemographic and Clinical Characteristics

The sociodemographic and clinical characteristics of the study population are shown in Table 2. A total of 727 individuals with TB/HIV co-infection were registered in the National TB Registry; 34 (4.6%) were the same individuals who had experienced more than one episode of TB, and therefore were excluded. Among the remaining 693 individuals, the majority (442; 58%) came from two sites (Chui and Bishkek) of the Kyrgyz Republic (Figure 1), 624 (90%) individuals had pulmonary TB, 197 (28%) individuals had previously treated TB, and 176 (25%) individuals had MDR-TB/XDR-TB.

The vast majority of all individuals (92%) received ART and CPT.

### 3.2. Key Population Groups

Eighty-four percent (84%) of the cohort had affiliations to one or more of eight different key population groups, with 49% of the cohort affiliated with two or more of such groups (Table 3).

### 3.3. TB Treatment Outcomes Stratified by Key Population Groups and Type of TB Drug Resistance

The TB treatment outcomes stratified by key population groups are shown in Table 4. Among all TB/HIV co-infected patients (N-693), 406 (59%) patients had successful treatment outcomes and 287 (41%) patients had unsuccessful outcomes. Unsuccessful outcomes significantly increased to 47% when there were ≥3 risk factors compared with 36% when there was only TB/HIV (*p* value-0.03).

Overall, treatment success for individuals with drug-sensitive TB was 63% and this declined to 50% with MDR-TB and to 6% with XDR-TB (Table 5). Death and loss to follow-up constituted the main components of unsuccessful treatment outcome, and there were only two individuals (<1%) with unevaluated outcomes.

### 3.4. Factors Associated with Unsuccessful Treatment Outcomes

After adjusting for potential confounders, significant risk factors associated with unsuccessful TB treatment outcomes included co-morbidities (including diabetes mellitus and hepatitis B/C), migration, alcohol usage, not being initiated on ART, and extrapulmonary TB (Table 6).

## 4. Discussion

The results of this first country-wide study showed that the great majority of individuals co-infected with TB/HIV had affiliations to one or more key population groups. Unsuccessful TB treatment outcomes increased by 11%, i.e., from 36% (TB/HIV alone) to 47%, when individuals with three or more key population affiliations came into play. Risk factors for unsuccessful outcomes included co-morbidities (diabetes mellitus and hepatitis B/C), migration, alcohol usage, not being on ART, and extrapulmonary TB.

From an operational perspective, the multiple key population affiliations are important as each group differs in their need for tailored access to clinical, social, and outreach services, thereby increasing the burden on the routine TB/HIV program. Similar cohort profiles with multiple affiliations have been reported by Tajikistan [15] and Kazakhstan [16].

The increase in unsuccessful TB outcomes (11%) when TB/HIV multiple key populations also come into play is a proxy of the challenge facing the existing TB/HIV services to provide adjunctive clinical and social support services, i.e., the health system is not designed to cope with the additional load. Another study from the Kyrgyz Republic also highlighted the higher risks of unsuccessful TB outcomes when there was an affiliation with key population groups [7].

This study is of national and regional significance. Meeting the end TB targets of providing 90% access to TB treatment and 90% treatment success for key populations [5] needs accelerated funding and support to put in place, effective, integrated, and holistic care. We recommend bridging the rhetoric with action by strengthening health system resilience in low- and-middle-income countries such as the Kyrgyz Republic. This will need a combined effort to mobilize funds from donors and to implement support by government, non-governmental organizations, and the community.

This study’s strengths are that we used country-wide data which were representative of the ground situation and that all treatment outcomes were cross validated and the proportion of unevaluated outcomes was negligible (<1%). This reflects a good performing monitoring system. We also adhered to STROBE guidelines for the reporting of observational studies in epidemiology [17]. In addition, the research topic is a recognized operational priority in the Kyrgyz Republic which increases the likelihood of influencing policy and practice change.

This study’s limitations are that data on diabetes mellitus and hepatitis B/C were aggregated together into one variable in the database. Since both morbidities are managed differently, this variable should be separated into different variables to allow deciphering and tailored management. This recommendation is already being taken up by the monitoring and evaluation team of the NTP. We also excluded 34 individuals from the analysis who experienced recurrent TB episodes during the study period, and although this is an important observation, it merits further specific research.

This study’s findings have programmatic implications in terms of management of TB/HIV in key populations. The following three key questions come to mind: (1) Where do we start? (2) Who should we target? (3) How does one move to country-wide implementation in the Kyrgyz Republic?

*Where do we start?* Since Bishkek and Chui regions had the highest burden of key populations and these regions have relatively better access to specialized services, it would make sense to start here. As experience is gained, progressively scaling up to other regions can be envisaged. As one paves the way forward, effective linkages will need to be built and gaps in clinical and support services bridged.

*Who should be targeted?* To get a “big bang for the buck”, those at risk for unsuccessful outcomes should be prioritized. They include co-morbidities (diabetes mellitus and hepatitis B/C), migrants/displaced, alcohol users, not being on ART, and those with extrapulmonary TB. Diabetes mellitus adversely affects TB treatment outcomes and increases the incidence of recurrent TB. Two systematic reviews involving 127 studies showed increased pooled relative risk of death of 1.89 (95% CI 1.5–2.4) in one review [18] and 1.88 (95% CI 1.6–2.2) in the other review [19]. The risk of TB relapse was 3.89 (95% CI 2.43–6.23) [18].

Hepatitis B/C are recognized as syndemics with HIV and TB, and they increase the risk of hepatotoxicity and worse outcomes when managed jointly with TB and ART treatment [20]. Migrants and displaced persons are often mobile and losses to follow-up and mortality are higher than in non-migrants [21]. Alcohol use is associated with worse treatment outcomes due to behavioral mechanisms, worse medication adherence, or biologic mechanisms including induction of liver enzymes that results in increased breakdown of rifampicin [22]. Finally, individuals who are HIV positive with extrapulmonary TB are in advanced clinical stages of HIV disease with a higher risk of death and so, too, those who are not on ART [23]. Targeted attention to and fast tracking of such groups is, thus, merited.

An encouraging finding was that the TB/HIV program managed to ensure that 92% of individuals in the cohort were placed on ART and a similar proportion were placed on cotrimoxazole preventive therapy. This is higher than what has been reported from Kazakhstan where ART uptake was 81% [16]. It was feasible to achieve this high uptake despite being affiliated to key populations. This shows that with adequate political and operational impetus patients who need adjunctive therapies could similarly be taken on board in the routine program setting [24].

*How to move towards country-wide implementation?* Moving forward for country-wide implementation to improve care would need to follow the following stepwise approach: (a) First, a country-wide mapping would need to be performed to identify facilities that could provide adjunctive clinical, social, and outreach services. Gap areas that need to be strengthened would be highlighted. (b) Second, health staff would need to be trained on referral and management procedures including the provision of adjunctive therapies. (c) Third, human and financial resources would need to be mobilized to ensure effective and holistic patient-centered care. (d) Fourth, important key population groups must be involved in the conception, design, and implementation of programs along the lines of “nothing for or about us, without us” [24].

## 5. Conclusions

In conclusion, for a long time, TB/HIV co-infected individuals have been recognized as a “double priority” for improving TB and HIV outcomes. Affiliation to key population groups accentuates their status to a “triple priority”.

In line with the sustainable development goal motto of “leave none behind” [3], we advocate for increased attention and equity [25] towards these vulnerable and hard to reach populations.

## Figures and Tables

**Figure 1 tropicalmed-08-00342-f001:**
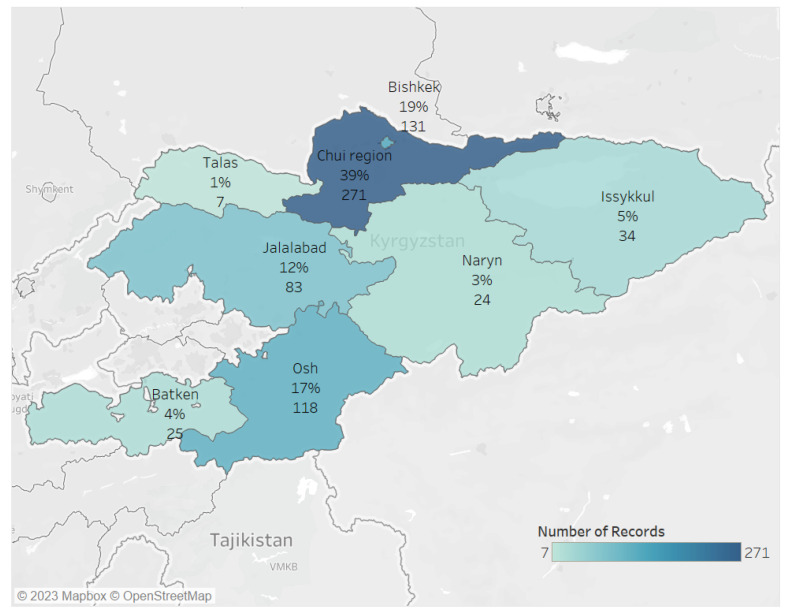
Map of the Kyrgyz Republic and the distribution of TB/HIV patients by regions (January 2018–June 2022).

**Table 1 tropicalmed-08-00342-t001:** Definitions of tuberculosis (TB) treatment outcomes.

Outcome	Definition
Cured	A pulmonary TB patient with bacteriologically confirmed TB at the beginning of treatment who completed treatment as recommended by the national policy, with evidence of bacteriological response and no evidence of failure.
Treatment completed	A patient who completed treatment as recommended by the national policy, whose outcome does not meet the definition for cure or treatment failure.
Treatment failed	A patient whose treatment regimen needed to be terminated or permanently changed to a new regimen or treatment strategy.
Died	A patient who died before starting treatment or during the course of treatment.
Lost to follow-up	A patient who did not start treatment or whose treatment was interrupted for 2 consecutive months or more.
Not evaluated	A patient for whom no treatment outcome was assigned.

**Table 2 tropicalmed-08-00342-t002:** Sociodemographic and clinical characteristics of TB/HIV co-infected patients in the Kyrgyz Republic, January 2018–June 2022.

Characteristics	Total N = 693	
	n	(%)
** *Sociodemographic* **		
Age (years)		
0–17	14	(2)
18–35	179	(26)
36–55	441	(64)
Gender		
Male	485	(70)
Female	208	(30)
Regions		
Chui	271	(39)
Bishkek	131	(19)
Osh	118	(17)
Jalalabad	83	(12)
Issykkul	34	(5)
Batken	25	(4)
Naryn	24	(4)
Talas	7	(1)
** *Clinical* **		
Type of TB		
Pulmonary	624	(90)
Extrapulmonary	69	(10)
History of TB treatment		
New	496	(72)
Previously treated	197	(28)
Smear status		
Positive	427	(62)
Negative	266	(38)
Drug resistance		
Sensitive	429	(62)
Poly-resistant	88	(13)
MDR/RR	160	(23)
Extensive drug resistant	16	(2)
ART		
Yes	635	(92)
No	54	(8)
Unknown	4	(1)
CPT		
Yes	634	(92)
No	59	(9)

ART, antiretroviral treatment; CPT, co-trimoxazole preventive therapy; HIV, human immunodeficiency virus; MDR/RR, multidrug- and rifampicin-resistant TB; TB, tuberculosis.

**Table 3 tropicalmed-08-00342-t003:** Key population group affiliations of TB/HIV co-infected patients in the Kyrgyz Republic, January 2018–June 2022.

Characteristics	Total N = 693	
	n	(%)
**Key population groups ^1^**		
Only TB/HIV co-infection	109	(16)
Unemployed	335	(48)
Co-morbidities (diabetes mellitus and hepatitis B/C)	244	(35)
Smokers	193	(28)
Migrants and displaced	121	(18)
Alcohol users	114	(17)
Ex-prisoners	90	(13)
Homeless	70	(10)
Intravenous drug users	72	(10)
**Affiliation to key populations**		
Only TB/HIV co-infection	109	(16)
1 key population	248	(36)
2 key populations	153	(22)
3 or more key populations	184	(27)

HIV, human immunodeficiency virus; TB, tuberculosis. ^1^ Each patient might have >1 risk factor, i.e., belong to >1 key population group, and therefore percentages do not add up to 100.

**Table 4 tropicalmed-08-00342-t004:** Treatment outcomes of TB/HIV patients stratified by key population groups in Kyrgyz Republic, January 2018–June 2022.

	Affiliation to a Key Population Groups									
	No Key Population ^1^		1 Key Population		2 Key Populations		≥3 Key Populations		Total	
	n	(%)	n	(%)	n	(%)	n	(%)	n	(%)
**Total**	108	(100)	248	(100)	153	(100)	184	(100)	693	(100)
**Successful outcomes**	69	(64)	149	60)	90	(59)	98	(53)	406	(59)
Cured	24	(22)	49	(20)	33	(22)	53	(29)	159	(23)
Completed	45	(42)	100	(40)	57	(37)	45	(24)	247	(36)
**Unsuccessful outcomes**	39	(36)	99	(40)	63	(41)	86	(47)	287	(41)
Died	24	(22)	59	(24)	35	(23)	46	(25)	164	(24)
Failure	1	(1)	11	(4)	6	(4)	6	(3)	24	(4)
Loss to follow up	14	(13)	28	(11)	22	(14)	34	(19)	98	(14)
Not evaluated	0	(0)	1	(<1)	0	(0)	0	(0)	1	(<1)

HIV, human immunodeficiency virus; TB, tuberculosis. ^1^ This group had only TB/HIV without being affiliated to any other key population group.

**Table 5 tropicalmed-08-00342-t005:** Treatment outcomes of TB/HIV patients stratified by drug sensitivity and key population groups in Kyrgyz Republic, January 2018-June 2022.

	Affiliation to Key Populations									
	No Key Population ^1^		1 Key Population		2 Key Populations		≥3 Key Populations		Total	
	n	(%)	n	(%)	n	(%)	n	(%)	n	(%)
Drug sensitive TB	69	(100)	169	(100)	88	(100)	103	(100)	429	(100)
Cured	8	(11)	28	(16)	15	(17)	28	(27)	79	(18)
Completed	40	(58)	86	(50)	39	(44)	28	(27)	193	(45)
Died	15	(21)	38	(22)	22	(25)	26	(25)	101	(23)
Failure	1	(1)	4	(2)	3	(3)	5	(4)	13	(3)
Loss to follow up	5	(7)	13	(7)	9	(10)	16	(15)	43	(10)
Not evaluated	0	(0)	0	(0)	0	(0)	0	(0)	0	(0)
Poly-resistant TB	17	(100)	21	(100)	20	(100)	30	(100.0)	88	(100.0)
Cured	7	(41)	7	(33)	8	(40)	14	(47)	36	(41)
Completed	4	(24)	5	(24)	4	(20)	3	(10)	16	(18)
Died	0	(0)	5	(24)	3	(15)	8	(27)	16	(18)
Failure	0	(0)	2	(9)	0	(0)	0	(0)	2	(2)
Loss to follow up	6	(35)	2	(9)	5	(25)	5	(17)	18	(21)
Not evaluated	0	(0)	0	(0)	0	(0)	0	(0)	0	(0)
Multidrug-resistant TB	19	(100)	53	(100)	41	(100)	47	(100)	160	(100)
Cured	9	(47)	14	(26)	10	(24)	11	(23)	44	(27)
Completed	1	(5)	8	(15)	14	(34)	14	(30)	37	(23)
Died	7	(37)	14	(26)	9	(22)	8	(17)	38	(24)
Failure	0	(0)	5	(9)	2	(5)	1	(2)	8	(5)
Loss to follow up	2	(11)	12	(23)	6	(15)	13	(28)	33	(21)
Not evaluated	0	(0)	0	(0)	0	(0)	0	(0.0)	0	(0)
Extensively drug-resistant TB	3	(100)	5	(100)	4	(100)	4	(100)	16	(100)
Cured	0	(0)	0	(0)	0	(0)	0	(0)	0	(0)
Completed	0	(0)	1	(20)	0	(0)	0	(0)	1	(6)
Died	2	(67)	2	(40)	1	(25)	4	(100)	9	(56)
Failure	0	(0)	0	(0)	1	(25)	0	(0)	1	(6)
Loss to follow up	1	(33)	1	(20)	2	(50)	0	(0)	4	(25)
Not evaluated	0	(0)	1	(20)	0	(0)	0	(0)	1	(6)

HIV, human immunodeficiency virus; TB, tuberculosis. ^1^ This group had only TB/HIV without being affiliated to any other key population group.

**Table 6 tropicalmed-08-00342-t006:** Risk factors associated with unsuccessful treatment outcomes among TB/HIV patients in the Kyrgyz Republic, January 2018–June 2022.

	Total	Unsuccessful Outcome		Crude RR (95% CI)	Adjusted RR (95% CI)	*p-*Value ^1^
	N	N	(%)			
Key population groups ^2^						
Unemployed						
No	358	158	(44)	Ref	Ref	
Yes	335	129	(39)	0.8 (0.7–1.0)	0.9 (0.7–1.1)	0.32
Co-morbidities						
No	449	174	(39)	Ref	Ref	
Yes	244	113	(46)	1.2 (1.0–1.4)	1.2 (1.0–1.4)	**0.02**
Smokers						
No	500	212	(42)	Ref	Ref	
Yes	193	75	(39)	0.9 (0.7–1.1)	0.8 (0.7–1.0)	0.06
Migrants and displaced						
No	572	226	(40)	Ref	Ref	
Yes	121	61	(50)	1.3 (1.0–1.6)	1.4 (1.2–1.7)	**0.001**
Alcohol users						
No	579	228	(38)	Ref	Ref	
Yes	114	59	(52)	1.3 (1.1–1.6)	1.3(1.0–1.7)	**0.02**
Ex-prisoners						
No	603	251	(42)	Ref	Ref	
Yes	90	36	(40)	0.9 (0.7–1.3)	0.9 (0.7–1.3)	0.84
Homeless						
No	623	250	(40)	Ref	Ref	
Yes	70	37	(53)	1.3 (1.0–1.7)	1.2 (0.9–1.6)	0.09
Intravenous drug users						
No	621	253	(41)	Ref	Ref	
Yes	72	34	(47)	1.2 (0.9–1.5)	1.0 (0.8–1.4)	0.86
Age (years)						
0–17	14	1	(7)	Ref	Ref	
18–55	620	261	(42)	5.9 (0.8–39)	5.1 (0.7–35.4)	0.74
56 and above	59	25	(42)	5.9 (0.8–40)	4.9 (0.7–34.2)	0.70
Gender						
Female	208	75	(36.1)	Ref	Ref	
Male	485	212	(43.7)	1.2 (0.9–1.5)	1.2 (0.9–1.4)	0.09
Type of TB						
Pulmonary	624	256	(41.0)	Ref	Ref	
Extra-pulmonary	69	31	(44.9)	1.1 (0.8–1.4)	1.4 (1.0–1.8)	**0.02**
History of TB treatment						
New	496	193	(38.9)	Ref	Ref	
Previously treated	197	94	(47.7)	1.2 (1.0–1.5)	1.1 (0.9–1.3)	0.41
ART						
Yes	635	229	(36)	Ref	Ref	
No	54	54	(100)	2.8 (2.5–3.1)	2.7 (2.3–3.1)	**<0.001**
CPT ^3^						
Yes	634	228	(36)	Ref	-	
No	59	59	(100)	2.8 (2.5–3.1)	-	**-**
TB drug resistance						
Drug-sensitive	429	157	(36.6)	Ref	Ref	
Poly-resistant	88	36	(40.9)	1.1 (0.8–1.5)	0.9 (0.7–1.3)	0.96
MDR-TB/XDR-TB	176	94	(53.4)	1.5 (1.2–1.8)	1.1 (0.9–1.4)	0.16

MDR/TB, multidrug-resistant TB; XDR-TB, extensively drug-resistant TB; ART, antiretroviral therapy; CPT, co-trimoxazole preventive therapy; HIV, human immunodeficiency virus; TB, tuberculosis; RR, relative risk. ^1^ Statistically significant *p*-values (*p* < 0.05) are shown in bold. ^2^ For each individual key population group (exposure), the combinations of all other groups were compared (unexposed). ^3^ CPT was dropped from adjusted analysis as it was highly collinear with ART.

## Data Availability

Requests to access these data should be sent to the corresponding author.

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
