# Peer review of "Triple Priority: TB/HIV Co-Infection and Treatment Outcomes among Key Populations in The Kyrgyz Republic: A National Cohort Study (2018–2022)"

_tropicalmed, 2023, doi:10.3390/tropicalmed8070342_

Round 1

Reviewer 1 Report

Suggestion for title: TB/HIV coinfection and treatment outcomes among key populations in the Kyrgyz Republic: A national TB program cohort study

Line 45: suggest  biological or behavioral factor to clinical and behavioral factors Line 53: consider : ensuring effective case finding and treatment.

In Section management: was short course TB treatment offered for drug resistant TB - what was the main ART regimen proposed to TB patients ?

In the statistical analysis section: please, mention if you have any diagnostic test for the multivariable logistic regression model

Consider line 163: consider replacing risk factor analysis by  multivariable logistic regression analysis.

Line 174:  suggestion: The vast majority of individuals (92%) was on ART and CPT

Table 1: Could you provide a rational about the age grouping? the intervals are different specifically from the third age group.

Discussion section: line 230: consider:  when individuals  with three or more key population affiliations came into play.

Line 237 increase instead of jump

line 238:  multiple key population affiliations

Please consider discussions your results as compared to similar work in neighboring countries. Most of the references are from or on Kyrgyz.

A  round of editing and proofreading will greatly improve the manuscript.

Reviewer 2 Report

This manuscript provides valuable descriptive information of treatment outcomes of TB patients living with HIV.

 Comments:

1.       The title emphasizes the role of “key populations” in the treatment responses of TB patients living with HIV. However, the authors only provided description information between key populations and successful outcomes in table 3. Please perform analyses to examine the association between being key populations and treatments outcomes.

2.       Please provide the rationale of using a “log binomial” model, but not a logistic regression model. Log binomial is not frequently used in epidemiological field.

3.       Line 159 said “P-value cutoff of 0.2 was used to select variables to be initially included in the regression model. However, in table 5, some variables which were likely to have a p-value>0.2 in the univariate analyses entered the multivariate analyses.

4.       In table 5, History of TB treatment, either the N in column 2 and 3 or the proportions on column 4 were wrong.

5.       In table 5, “Key population groups”. It is not clear how this part was done. First, there was no reference group. Second, key population groups are not mutually exclusive. Please check and revise.

6.       In table 5, some of the reference groups are likely mislabeled. E.g. the risk of having unsuccessful treatment outcome for patients who were under ART was obviously higher than those who were not, but the RR for No-ART v. Yes-Art was adverse.

7.       The cutoffs for age variable were a bit unusual. Please provide rationales or revise.

8.       In table 5, it suggests that all patients who were under ART or CPT had unfavorable TB treatment outcomes. Please first confirm this information is correct. If it does, please emphasize and discuss this finding in the discussion section.

This article is clear and understandable. However, please see English editor for minor revision.

Round 2

Reviewer 2 Report

In the response 6, you mentioned that there is no specific reference category. This may work for the univariate analyses, but how it worked for the multivariate adjusted model? Please further clarify.  

Author Response

POINT BY POINT RESPONSE TO THE REVIEWER 2

We thank the reviewer for reviewing this paper once again and for the comment related to response 6.

The specific suggestion is highlighted in bold font and our response in blue font

Comment: In the response 6, you mentioned that there is no specific reference category. This may work for the univariate analyses, but how it worked for the multivariate adjusted model? Please further clarify.  

Response: For each individual key population group (exposure), the combination of all other groups are compared (unexposed). We have now included reference categories against each key population group in Table 5 for better clarity. This makes the reference categories clear. We hope this addresses the reviewer’s concern.
